# Classical and Alternative Pathways of the Renin–Angiotensin–Aldosterone System in Regulating Blood Pressure in Hypertension and Obese Adolescents

**DOI:** 10.3390/biomedicines12030620

**Published:** 2024-03-10

**Authors:** Adrian Martyniak, Dorota Drożdż, Przemysław J. Tomasik

**Affiliations:** 1Department of Clinical Biochemistry, Institute of Pediatrics, Jagiellonian University Medical College, 30-663 Krakow, Poland; adrian.martyniak@uj.edu.pl; 2Department of Pediatric Nephrology and Hypertension, Institute of Pediatrics, Jagiellonian University Medical College, 30-663 Krakow, Poland; dorota.drozdz@uj.edu.pl

**Keywords:** angiotensin II, angiotensin 1–7, angiotensin IV, angiotensin 1–9, obesity, arterial hypertension

## Abstract

Primary hypertension (PH) is the leading form of arterial hypertension (AH) in adolescents. Hypertension is most common in obese patients, where 20 to 40% of the population has elevated blood pressure. One of the most effective mechanisms for regulating blood pressure is the renin–angiotensin–aldosterone system (RAAS). The new approach to the RAAS talks about two opposing pathways between which a state of equilibrium develops. One of them is a classical pathway, which is responsible for increasing blood pressure and is represented mainly by the angiotensin II (Ang II) peptide and, to a lesser extent, by angiotensin IV (Ang IV). The alternative pathway is responsible for the decrease in blood pressure and is mainly represented by angiotensin 1–7 (Ang 1–7) and angiotensin 1–9 (Ang 1–9). Our research study aimed to assess changes in angiotensin II, angiotensin IV, angiotensin 1–7, and angiotensin 1–9 concentrations in the plasma of adolescents with hypertension, with hypertension and obesity, and obesity patients. The Ang IV concentration was lower in hypertension + obesity versus control and obesity versus control, respectively *p* = 0.01 and *p* = 0.028. The Ang 1–9 concentration was lower in the obesity group compared to the control group (*p* = 0.036). There were no differences in Ang II and Ang 1–7 peptide concentrations in the hypertension, hypertension and obesity, obesity, and control groups. However, differences were observed in the secondary peptides, Ang IV and Ang 1–9. In both cases, the differences were related to obesity.

## 1. Introduction

Cardiovascular disease (CVD) is the leading cause of death in the world. The WHO report (2021) estimated 17.9 million deaths each year [1]. Only in the United States (USA), one in every five people dies of CVD, and heart disease costs the United States approximately USD 239.9 billion per year [2]. One of the main causes of CVD is hypertension. Arterial hypertension (AH) is a very serious disease that leads to many complications, including death, kidney failure, and myocardial infarction. Hypertension is a serious medical problem in adolescents and even in children. In the US, hypertension is estimated to be associated with 0.3 to 4.5% of the pediatric population [3]. Primary hypertension (PH) is the leading form of arterial hypertension in adolescents. Hypertension is more common in obese patients, where 20 to 40% of the population has elevated blood pressure [4]. In addition to obesity, genetic and environmental factors play an important role [4].

AH symptoms are nonspecific and often difficult to observe. Patients complain of headaches, fatigue, nosebleeds, sleep disturbances, and nervousness. Although the diagnostic criteria for hypertension in adults are well known and widespread, there is no such consensus among adolescents.

An important factor in hypertension is obesity. Although obesity is the underlying disease, the associated hypertension is still considered primary [4]. The effects of obesity on blood pressure are multiple and include hormonal, neurological, and anatomical changes. The most important are insulin resistance, increased sympathetic nervous system activity, increased renin release, increased circulating blood volume, and increased circulating blood resistance [5].

One of the most effective mechanisms to regulate blood pressure is the renin–angiotensin–aldosterone system (RAAS) [6]. The reaction cascade starts from the enzymatic fragmentation of angiotensinogen. Angiotensinogen is a peptide hormone, produced in the liver, that is a precursor of all angiotensin peptides [7]. Angiotensin peptides, despite a small molecular weight and similar structure, act in a different role. The RAAS causes the retention of water and sodium, the release of aldosterone, the contraction of blood vessels, increased heart rate, and consequently hypertension. On the other hand, the RAAS can decrease blood pressure through several reverse mechanisms, such as the vasodilation of blood vessels or the release of nitric oxide [8]. Therefore, the new approach to the RAAS talks about two opposing pathways between which a state of equilibrium develops. One of them is a classical pathway that is responsible for increasing blood pressure. The main effector of the classical pathway is angiotensin II (Ang II) and, to a lesser extent, angiotensin IV (Ang IV). The second pathway is called an alternative. The alternative pathway is responsible for decreasing blood pressure and is mainly represented by angiotensin 1–7 (Ang 1–7) and, to a lesser extent, angiotensin 1–9 (Ang 1–9) [9,10].

Our research study aimed to assess changes in angiotensin II, angiotensin IV, angiotensin 1–7, and angiotensin 1–9 concentrations in plasma of adolescents with hypertension, hypertension and obesity, and obese patients with normal blood pressure. Based on measured concentrations of the main peptides of classical and alternative RAAS pathways, the disbalance in these pathways in AH and obese adolescents was analyzed. A secondary aim was to identify a marker predictive of arterial hypertension.

## 2. Materials and Methods

We recruited adolescents suffering from AH, obesity, and combinations of these diseases among patients from the Department of Pediatrics Nephrology and Hypertension of the University Children’s Hospital in Krakow, Poland. All patients were recruited to this study by the hypertensiologist, according to the consensus of the European Society of Hypertension (ESH) named Pediatric Hypertension Guidelines 2016 [11], and related patient data were collected from their medical records. Patients with the E66 (ICD-10) diagnosis in medical documentation were eligible for the obese group. Fasting blood samples (S-Monovette EDTA K3E/2.6 mL, Sarstedt AG & Co.KG, Numbrecht, Germany) were taken in the morning, immediately cooled, and then centrifuged. Adolescents in the control group were recruited from families and friends of the study researchers. These adolescents had no pathological clinical signs or complaints or any pharmacological treatment. In the control group, fasting blood samples were drawn in the same procedure as in the study group.

Blood samples were centrifuged and separated, and EDTA plasma was frozen at −80 centigrade until measurement. The maximum bank loan time was not longer than 12 months. Peptide concentrations were measured using commercially available enzyme-linked immunosorbent assay (ELISA) immunoassays: angiotensin II assay range: 12.5 ng/mL–800 ng/mL, angiotensin IV assay range: 1.56 ng/mL–100 ng/mL, angiotensin 1–7 assay range: 12.5 ng/mL–800 ng/mL, and angiotensin 1–9 assay range: 7.8 ng/mL–500 ng/mL (Qayee Bio-Technology Co., Ltd., Shanghai, China). The manufacturer declares that there is no cross-reaction and coefficient variation < 15%. The samples were slowly defrosted. The first step was the transfer of the samples from −80 centigrade to −20 centigrade for a night; then, they were thawed in ice-free water. According to the manufacturer’s guidelines, all samples were diluted five times before analysis.

The assay procedures were performed according to the manufacturer’s manuals using a Bio-Rad washer and plate reader (Bio-Rad, Hercules, CA, USA). According to the manufacturer, all of the tests used have high sensitivity and excellent specificity for the detection of the measured parameters without significant cross-reactivity or interference between the analytes and their analogues. 

The study protocol was approved by the Jagiellonian University Bioethical Committee (approval no. 1072.61.20.67.2019), and informed consent was obtained from all the legal guardians of the patients and all the patients over 16 years of age enrolled in the study.

The statistical analysis was performed using IBM SPSS Statistics (v29, IBM Corporation, Armonk, NY, USA). The concentrations of angiotensin derivative peptides were expressed as median values and quartile ranges. Normality was checked using the Shapiro–Wilk test in each group. The Kruskal–Wallis test was performed for comparisons between the studied groups. If the Kruskal–Wallis test did not show any significant statistical differences, but the analysis of differences in individual groups showed possible differences, a U Mann–Whitney test was performed. 

## 3. Results

We studied 28 patients with AH (15.05 ys ± 2.98; BMI 21.75 ± 3.44 kg/m^2^), 17 patients with AH and obesity (13.95 ys ± 3.79; BMI 29.89 ± 4.64 kg/m^2^), and 29 patients with obesity (13.50 ys ± 3.39; BMI 28.40 ± 5.59 kg/m^2^) and normal blood pressure. As a control group, 52 healthy children were observed with normal BMI (12.95 ys ± 3.69; BMI 18.63 ± 3.9 kg/m^2^) and normal blood pressure. All patient characteristics are summarized in Table 1.

### 3.1. Angiotensin II

Plasma Ang II concentrations did not differ statistically in the analyzed groups. The median and quartiles 1 and 3, respectively, were 275.04 (245.89–344.33) ng/mL in hypertension, 285.12 (236.95–323.60) ng/mL in hypertension and obesity, and 281.11 (256.11–321.37) ng/mL in obesity. The concentration of Ang II in the control group was 308.05 (271.52–372.24) ng/mL. There were also no statistical differences between the study and the control group (Figure 1).

### 3.2. Angiotensin IV

Plasma Ang IV concentrations differed statistically in the analyzed groups (*p* = 0.017). The median and quartiles 1 and 3, respectively, were 34.51 (15.15–49.07) ng/mL in hypertension, 29.11 (19.07–38.98) ng/mL in hypertension and obesity, and 33.55 (24.52–40.71) ng/mL in obesity. The concentration of Ang IV in the control group was 39.71 (32.09–47.75) ng/mL. The significant statistical differences were between hypertension + obesity and control and obesity versus control, respectively *p* = 0.01 and *p* = 0.028 (Figure 2).

### 3.3. Angiotensin 1–7

Plasma Ang 1–7 concentrations did not differ statistically in the analyzed groups. The median and quartiles 1 and 3, respectively, were 302.11 (255.87–367.40) ng/mL in hypertension, 268.01 (225.16–345.57) ng/mL in hypertension and obesity, and 283.42 (237.50–341.61) ng/mL in obesity. The concentration of Ang 1–7 in the control group was 268.76 (236.29–380.56) ng/mL. There were no statistical differences between the study and the control group (Figure 3).

### 3.4. Angiotensin 1–9

Plasma Ang 1–9 concentrations did not differ statistically in the analyzed groups. The median and quartiles 1 and 3, respectively, were 181.38 (159.69–205.30) ng/mL in hypertension, 169.16 (147.68–203.13) ng/mL in hypertension and obesity, and 172.07 (155.81–205.67) ng/mL in obesity. The concentration of Ang 1–9 in the control group was 193.30 (169.23–215.63) ng/mL. A statistical difference between the obesity group and the control group (*p* = 0.036) was confirmed by the U Mann–Whitney test (Figure 4).

## 4. Discussion

Several studies describe the concentrations of chosen angiotensin peptides. However, due to differences in the measurement methods, it is difficult to compare the subsequent peptide levels of different papers. This study is the first to comprehensively present and analyze both RAAS pathways in adolescents with hypertension.

This study is innovative for several reasons. It is the first study to comprehensively analyze the classical and alternative RAA systems in the adolescence period (10–18 years). According to Litwin et al., this is when idiopathic hypertension occurs most often. Authors analyze the RAA system as two separate pathways that should be in balance. The disbalance of these pathways may result in AH. Some angiotensin peptides such as Ang IV and Ang 1–9 have never been previously investigated as potential factors involved in the development of AH.

Angiotensin II (Asp-Arg-Val-Tyr-Ile-His-Pro-Phe) has the strongest biological activity in the RAAS classical pathway. The peptide has a strong affinity for the angiotensin type I receptor (AT1R). The AT1R is located mainly in the kidneys, vascular smooth muscle, lungs, and liver. The receptor belongs to the superfamily of G protein-bound receptors. The stimulation of the AT1R affects the release of aldosterone from the adrenal cortex, vasoconstriction, the activation of inflammatory processes, fibrosis, and myocardial hypertrophy [12,13]. In 2004, Silva et al. compared several angiotensin derivatives in children and adolescents with hypertension and normotension (3.1–16.7 ys). Similar to our study, the concentration of Ang II was almost identical in both groups (21.4 ± 8.7 vs. 22.2 ± 10.3 pg/mL [RIA]) [14]. Also, Al-Daghri et al., in 2010, compared Ang II concentrations in lean and obese children and adolescents (5–12 years). In this study, the researchers did not show a difference between the concentration of Ang II in lean and obese patients (0.70 ± 0.32 vs. 0.51 ± 0.13 boys and 0.65 ± 0.33 vs. 0.95 ± 1.0 girls) or between boys and girls (0.65 ± 0.3 vs. 0.73 ± 0.6) [15]. The results obtained in our study are compatible with previous studies. These results seem to suggest the existence of a local renin–angiotensin system. For many years, the RAAS has been described as a systemic system. This local system may influence blood pressure, while peripheral concentrations of angiotensin derivatives are at normal levels. The local renin–angiotensin system (RAS) is present in many tissues and organs such as muscle, the heart, the nervous system, bone, the kidneys, and the brain [9].

Angiotensin IV (Val-Tyr-Ile-His-Pro-Phe) has low biological activity. Ang IV has an affinity for the type IV angiotensin receptor (ATR4). The ATR4 receptor is widely distributed and is found in many tissues such as the brain, adrenal glands, kidneys, lungs, and heart. The ATR4 receptor is a transmembrane enzyme, insulin-regulated membrane aminopeptidase (IRAP). In the kidneys, Ang IV increases blood flow and decreases the transport of sodium ions to the proximal tubules in the kidney. However, the effect of arterial blood pressure on Ang IV is minimal. Ang IV may also have an influence on mental disorders. By regulating blood flow in the structure of the brain, Ang IV participates in neural plasticity, learning, and memory processes [16,17].

There are no studies that describe the concentration of Ang IV in the plasma of children and adolescents. Many studies described Ang IV as a very important brain blood regulatory peptide [18,19]. Animal studies, particularly rats, confirm the hypertensive effect of Ang IV [20]. However, due to the structure of the molecule and the activity of endopeptidases, the peripheral activity of Ang IV may be limited. However, the confirmation of this effect requires additional research, in particular on the local RAS. Ang IV and the local RAS play an important role in obesity and insulin resistance. In 2011, Wang et al. conducted studies in rats, where they showed that Ang IV causes an increase in glucose tolerance and insulin signaling [21]. The local RAS in adipose tissue has a significant impact on the development of obesity. The presence of Ang IV increases glucose uptake [22]. In our study, the concentration of Ang IV was significantly lower in the obesity and hypertension + obesity groups. This direction is in line with the expectations.

Angiotensin 1–7 (Asp-Arg-Val-Tyr-Ile-His-Pro) is formed directly from Ang II and, to a lesser extent, from different angiotensin derivatives. Ang 1–7 is the selective endogenous ligand for Mas receptors (MasRs). The MasR belongs to the G protein-coupled receptor family. The MasR is located in the endothelium of blood vessels, macrophages, and neurones. Ang 1–7 decreases blood pressure due to the vasodilation of blood vessels, the synthesis of anti-inflammatory prostaglandins, and the release of nitric oxide. Ang 1–7 also has a positive effect on the myocardium, limiting hypertrophy and cardiomyocyte proliferation. In the kidneys, Ang 1–7 regulates sodium ion transport and increases glucose resorption [23,24]. 

There are a few studies that describe the concentration of Ang 1–7 in children and adolescents. Silva et al., in the same study as Ang II, determined Ang 1–7 concentrations. Patients with essential hypertension have significantly higher Ang 1–7 concentrations than in the control group (78.8 ± 22.8 vs. 16.2 ± 7.9 [pg/mL] *p* < 0.05) [14]. Our results are in opposition to the work of Silva et al. because in our study, there are no differences between the hypertension and control groups, similar results to those received by Kohara et al. In their studies on hypertensive rats, they obtained a 3.7 times higher concentration of Ang 1–7 than in healthy rats (*p* < 0.05) [25]. Cambell et al. obtained different results. In their study, plasma concentrations of Ang 1–7 were similar in hypertensive and normotensive rats [26]. Large discrepancies in the test results may be the result of the in vivo and ex vivo degradation of the peptide. The authors of the studies used methods to prevent enzymatic degradation, such as cooling the sample and/or using inhibitors. The tissue RAS and related proteolytic enzymes are also of importance. They identify at least a few enzymes involved in Ang 1–7 degradation, such as angiotensin-converting enzyme 2 (ACE2), decarboxylase, aminopeptidase, or angiotensin-converting enzyme (ACE) [27]. From a clinical point of view, these differences are difficult to describe. Perhaps the increased concentration of the peptide is due to an attempt to achieve balance in the RAAS. Shifting the balance towards the alternative axis and lowering Ang II to Ang 1–7 could contribute to lower blood pressure.

Angiotensin 1–9 (Asp-Arg-Val-Tyr-Ile-His-Pro-Phe-His) has been found in many tissues and organs, such as the heart, kidneys, and testes. The highest concentrations have been observed in the endothelium of the coronary vessels. Ang 1–9 demonstrates a very similar effect to Ang 1–7. Ang 1–9 has a stronger effect on the myocardium than on the blood vessel. Angiotensin 1–9 acts through the angiotensin receptor type II (AT2R) [28].

No studies have been found that describe the concentration of Ang 1–9 in hypertensive children and adolescents. There are several studies in rats that confirm the hypotensive effect of Ang 1–9 [29,30,31]. Similarly to Ang IV, Ang 1–9 may play an important role in the local RAS. Due to the relatively high degradation of the peptide to Ang 1–7, it plays a secondary role. However, the reduced concentration in the group of obese patients suggests the influence of adipose tissue on the peripheral concentration of Ang 1–9 Perhaps ACE2-rich adipose tissue degrades large amounts of Ang 1–9 to Ang 1–7, shifting the balance of the RAAS toward cardioprotective and hypotensive effects. However, this requires further research.

## 5. Conclusions

The renin–angiotensin–aldosterone system is an extremely complex mechanism. The RAAS consists of many peptides and enzymes. In the system and among the measured peptides, Ang II and Ang 1–7 have the strongest biological activity. In this study, there are no differences in the concentrations of those peptides in the hypertension, hypertension and obesity, obesity, and control groups. However, differences are observed in the secondary peptides Ang IV and Ang 1–9. These peptides could be a predictor of AH in obese patients, but this hypothesis should be confirmed by prospective studies. This may mean that even minor disturbances in the RAAS of homeostasis lead to hypertension. On the other hand, the classical pathway of the RAAS system appears to be resistant to changes in the organism, and the alternative axis may be responsible for the development of hypertension. The disbalance found in this study is shown in Figure 5.

A detailed understanding of the pathogenesis of hypertension and the impact of obesity requires further research. Only a small fraction of the RAAS was analyzed in our study. Among angiotensin derivatives, there are still many promising peptides that can be used in the diagnosis, monitoring, or even treatment of hypertension in children and adolescents. A very promising direction of development is research on Ang 1–7 and the MasR. Although Ang 1–7 is the only endogenous ligand for the MasR, its analogues are actively sought. The stimulation of the Mas receptor can bring many benefits to people with hypertension. In addition to vasodilation and NO synthesis, the MasR can induce the synthesis of anti-inflammatory prostaglandins. This is important in inflammatory processes and myocardial fibrosis. Therefore, the ACE2/Ang 1–7/MasR axis is a very promising development direction. Another conclusion of the study is the need to better understand local RAAs. Local RAAs perform very important functions in regulating blood flow and inflammatory processes. Unfortunately, its role in systemic action and clinical use is currently unknown. In the case of obese people, excessively developed adipose tissue is assumed to assume the role of the endocrine organ, which can affect the economy of the whole organism.

As shown in Figure 5, the balance of RAAS shifts to the alternative pathway in obese patients with hypertension and obese patients with normal blood pressure. Perhaps this is the result of compensatory mechanisms operating in the RAAS and/or that are associated with obesity. Maintained balance in patients with hypertension may indicate that compensation possibilities have been exhausted and other mechanisms are involved in the regulation of blood pressure.

## 6. Limitations of the Study

The main limitation of this study was the high biological and analytical variability of the analyzed compounds. Because the tested peptides were unstable in plasma, laboratory control over the pre-analytical phase and material storage were very important.

## Figures and Tables

**Figure 1 biomedicines-12-00620-f001:**
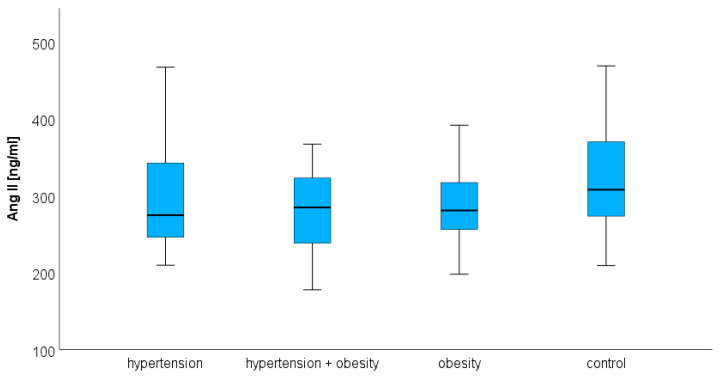
The median concentration of angiotensin II in the hypertension group, hypertension and obesity group, obesity group, and control group. The boxes show the median and quartile range of the measured plasma angiotensin II concentrations in the study and control group; the whiskers show the minimal and maximal measured concentration. No significant differences were observed.

**Figure 2 biomedicines-12-00620-f002:**
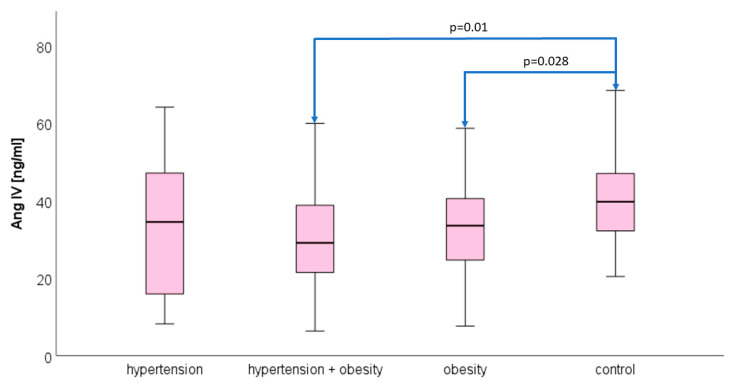
The median concentration of angiotensin IV in the hypertension group, hypertension and obesity group, obesity group, and control group. The boxes show the median and quartile range of the measured plasma angiotensin IV concentrations; the whiskers show the minimal and maximal measured concentration: a significant difference between adolescents with hypertension and the obesity vs. control group, *p* = 0.01; a significant difference between obese adolescents and the control group, *p* = 0.028.

**Figure 3 biomedicines-12-00620-f003:**
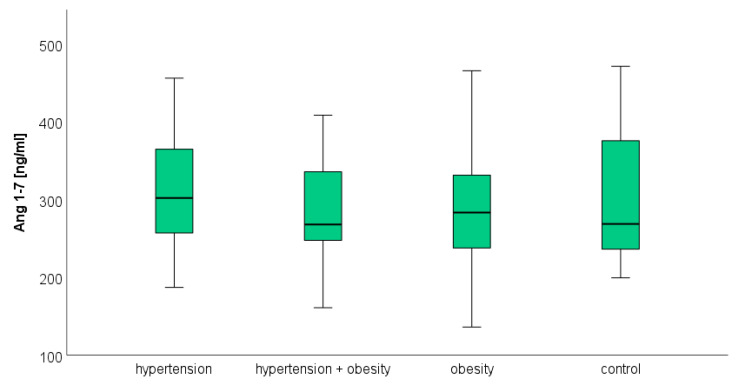
The median concentration of angiotensin 1–7 in the hypertension group, hypertension and obesity group, obesity group, and control group. The boxes show the median and quartile range of the measured plasma angiotensin 1–7 concentrations in the study and control group; the whiskers show the minimal and maximal measured concentration. No significant differences were observed.

**Figure 4 biomedicines-12-00620-f004:**
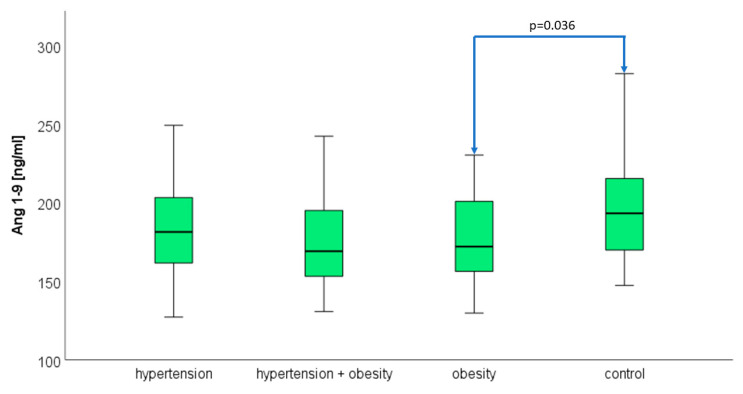
The median concentration of angiotensin 1–9 in the hypertension group, hypertension and obesity group, obesity group, and control group. The boxes show the median and quartile range of the measured plasma angiotensin 1–9 concentrations in the study and control group; the whiskers show the minimal and maximal measured concentration: a significant difference between obese adolescents and the control group was confirmed by the U Mann–Whitney test, *p* = 0.036.

**Figure 5 biomedicines-12-00620-f005:**
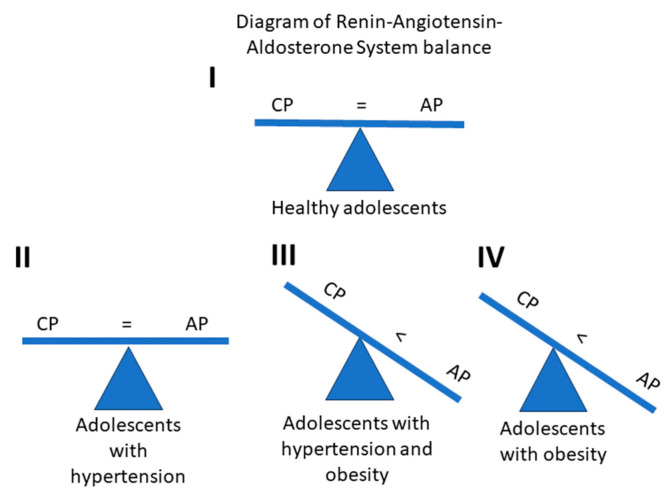
This diagram represents a theoretical balance between the classical pathway (CP) and alternative pathway (AP) of the renin–angiotensin–aldosterone system; (**I**) shows a fully balanced RAAS in healthy adolescents; (**II**) shows a theoretical balance in adolescents with hypertension, where the activity of the CP equals the activity of the AP; (**III**) shows a disbalance of the RAAS in obese adolescents with hypertension. The AP is more active than the CP; (**IV**) shows a disbalance of the RAAS in obese adolescents. The AP is more active than the CP.

**Table 1 biomedicines-12-00620-t001:** Patients’ characteristics.

	Number of Patients	Age	BMI	Systolic Pressure (SP)	Diastolic Pressure (DP)
Hypertension	28	15.05 ys ± 2.98	21.75 ± 3.44 kg/m^2^	132 ± 16 mmHg	79 ± 11 mmHg
Hypertension + obesity	17	13.95 ys ± 3.79	29.89 ± 4.64 kg/m^2^	138 ± 19 mmHg	74 ± 14 mmHg
Obesity	29	13.50 ys ± 3.39	28.40 ± 5.59 kg/m^2^	114 ± 10 mmHg	68 ± 9 mmHg
Control	52	12.95 ys ± 3.69	18.63 ± 3.9 kg/m^2^	112 ± 11 mmHg	66 ± 10 mmHg

## Data Availability

Data are contained within the article.

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
