# Peer review of "Classical and Alternative Pathways of the Renin–Angiotensin–Aldosterone System in Regulating Blood Pressure in Hypertension and Obese Adolescents"

_biomedicines, 2024, doi:10.3390/biomedicines12030620_

Round 1

Reviewer 1 Report

Comments and Suggestions for Authors

I am very confused by the statistical analysis.  The methods section describes use of the Kruskal-Wallis test, a non-parametric equivalent of ANOVA.  By convention if  K-W test or ANOVA determine that there are no significant differences between the groups of the study, that finding should stand.  For each of the peptides studied, the authors state that there were no differences in the concentrations of the peptide studies between the groups.  They then appear to go on to make comparisons between individual groups.  Such an approach is invalid and negates the use and findings of the K-W test.

I believe that the sample size is sufficient to detect any meaningful differences between groups, although this could be checked formally by test power analysis.

Based on the results shown I believe that this study indicates that there are no significant changes in angiotensin peptide concentrations between the groups studied.

Comments on the Quality of English Language

The English language is good, but clearly not written by a native English-speaking scientist (eg centrifugated rather than centrifuged).  The meaning is clear.  

Author Response

Sir/Madam

Thank you very much for the review and kind opinion.

I am very confused by the statistical analysis.  The methods section describes use of the Kruskal-Wallis test, a non-parametric equivalent of ANOVA.  By convention if  K-W test or ANOVA determine that there are no significant differences between the groups of the study, that finding should stand.  For each of the peptides studied, the authors state that there were no differences in the concentrations of the peptide studies between the groups.  They then appear to go on to make comparisons between individual groups.  Such an approach is invalid and negates the use and findings of the K-W test.

The KW test showed statistical significance for Ang IV and the p-score was included in the manuscript. In the case of Ang 1-9, where the KW test did not show significant differences (p>0.05), a different statistical test was used. Appropriate annotations have been included

I believe that the sample size is sufficient to detect any meaningful differences between groups, although this could be checked formally by test power analysis.

Thanks you for a valuable comment. The limitations was changed.

Based on the results shown I believe that this study indicates that there are no significant changes in angiotensin peptide concentrations between the groups studied.

Significant changes are confirmed in Ang IV

Reviewer 2 Report

Comments and Suggestions for Authors

Adrian et al. show that changes in the concentrations of angiotensin-II, angiotensin IV, angiotensin 1-7, and angiotensin 1-9 in the plasma of adolescents with hypertension, hypertension with obesity, and obesity patients. Multiple studies revealed these changes with their respective studies. The novelty of the study is questionable. 

Comments on the Quality of English Language

No issues with the english

Author Response

Sir/Madam

Thank you very much for the review and valuable opinion.

Adrian et al. show that changes in the concentrations of angiotensin-II, angiotensin IV, angiotensin 1-7, and angiotensin 1-9 in the plasma of adolescents with hypertension, hypertension with obesity, and obesity patients. Multiple studies revealed these changes with their respective studies. The novelty of the study is questionable. 

The novelty and originality of the research lies in the simultaneous analysis of both RAA pathways simultaneously based on defined peptides. The studied groups i.e. the age of patients are also rare in published studies on hypertension.

Reviewer 3 Report

Comments and Suggestions for Authors

This study raises several concerns:

1/ I do not understand the aim of this study.

It seems to me that authors want to analyze the role of several specific effectors of the RAAS on the occurrence of hypertension both in lean and obese subjects. Therefore, the comparisons should be between lean hypertensive subjects and lean controls on one hand, and obese hypertensive subjects versus obese controls on the other hand. It could be the primary aim.

If the aim is to analyze the relation between obesity and the levels of several RAAS effectors, the comparisons should be between obese controls and lean controls on one hand, and between obese hypertensive patients and hypertensive controls on the other hand. It could be a secondary aim.

2/ I do not understand the statistics. Did the authors compared each of the three disease groups to controls? In this case, authors should make correction for multiple comparisons. This is another argument in favor of my first comment.

3/ AngIV levels were lower in obese hypertensive patients compared to lean controls but not compared to obese controls. Furthermore, lean hypertensive subjects and lean controls had comparable AngIV levels, ruling out an impact of hypertension. Obese hypertensive and lean hypertensive subjects also had comparable AngIV levels, ruling out an impact of obesity.

Ang1-9 levels were lower in obese patients compared to lean controls but not compared to obese hypertensive patients. Furthermore, lean hypertensive subjects and lean controls had comparable Ang1-9 levels, ruling out an impact of hypertension. Obese hypertensive and lean hypertensive subjects also had comparable Ang1-9 levels, ruling out an impact of obesity.

Therefore, I fear that this study is in fact negative and do not allow to raise any significant conclusion.

Comments on the Quality of English Language

Extensive English editing is required

Author Response

Sir/Madam

Thank you very much for the review and insightful analysis.

This study raises several concerns:

1/ I do not understand the aim of this study.

It seems to me that authors want to analyze the role of several specific effectors of the RAAS on the occurrence of hypertension both in lean and obese subjects. Therefore, the comparisons should be between lean hypertensive subjects and lean controls on one hand, and obese hypertensive subjects versus obese controls on the other hand. It could be the primary aim.

These comparisons were made in a multiplex analysis. The aim of the study was clarified.

If the aim is to analyze the relation between obesity and the levels of several RAAS effectors, the comparisons should be between obese controls and lean controls on one hand, and between obese hypertensive patients and hypertensive controls on the other hand. It could be a secondary aim.

Also, the secondary aim was improved.

2/ I do not understand the statistics. Did the authors compared each of the three disease groups to controls? In this case, authors should make correction for multiple comparisons. This is another argument in favor of my first comment.

Yes, the study groups were compared to the control group and with each other. Correction for multiple comparisons are not mandatory in this type of research. Correction for multiple comparisons would only be indicated in the case of Ang IV (p<0.05). Appropriate annotation was added to the manuscript.

3/ AngIV levels were lower in obese hypertensive patients compared to lean controls but not compared to obese controls. Furthermore, lean hypertensive subjects and lean controls had comparable AngIV levels, ruling out an impact of hypertension. Obese hypertensive and lean hypertensive subjects also had comparable AngIV levels, ruling out an impact of obesity.

Ang1-9 levels were lower in obese patients compared to lean controls but not compared to obese hypertensive patients. Furthermore, lean hypertensive subjects and lean controls had comparable Ang1-9 levels, ruling out an impact of hypertension. Obese hypertensive and lean hypertensive subjects also had comparable Ang1-9 levels, ruling out an impact of obesity.

Therefore, I fear that this study is in fact negative and do not allow to raise any significant conclusion.

The authors reliably report the results and attempt a discussion based on available literature and confirmed medical knowledge. The conclusions drawn constitute a hypothesis and require confirmation, which has been clearly indicated.

Round 2

Reviewer 1 Report

Comments and Suggestions for Authors

I am still uncertain of the validity of undertaking a K-W test, ignoring the results, and then conducting multiple comparisons.  However appropriate information about the statistical procedures is included thus the reader can form their own opinions.

Author Response

Sir/Madam

Thank you very much for the second review.

I am still uncertain of the validity of undertaking a K-W test, ignoring the results, and then conducting multiple comparisons.  However appropriate information about the statistical procedures is included thus the reader can form their own opinions.

Thank you for your opinion. We agree with your comment. However, we believe that the statistical calculations and conclusions have been presented fairly and clearly and are worth presenting to the scientists. We also believe in the usefulness of the results in the design of a new pharmacological approach to hypertension.

Reviewer 2 Report

Comments and Suggestions for Authors

Please elaborate on the current study's novelty. still questionable. 

Author Response

Sir/Madam

Thank you very much for the second review.

Please elaborate on the current study's novelty. still questionable. 

Please, look at the top of the discussion. We added a few more sentences on the novelty of this study.

“The study is innovative for several reasons. It is the first study to comprehensively analyse the classical and alternative RAA system in the adolescence period (10-18 years). According to Litwin et al., this is when idiopathic hypertension occurs most often. Authors analyse the RAA system as the two separate pathways that should be in a balance. The disbalance of these pathways may result in AH. Some angiotensin peptides such as Ang IV and Ang 1-9 have never previously been investigated as potential factors involved in the development of AH. "